# Purple Wheat: Food Development, Anthocyanin Stability, and Potential Health Benefits

**DOI:** 10.3390/foods12071358

**Published:** 2023-03-23

**Authors:** Tamer H. Gamel, Syed Muhammad Ghufran Saeed, Rashida Ali, El-Sayed M. Abdel-Aal

**Affiliations:** 1Guelph Research and Development Centre, Agriculture and Agri-Food Canada, 93 Stone Road West, Guelph, ON N1G 5C9, Canada; 2Department of Food Science and Technology, University of Karachi, Karachi 75270, Pakistan

**Keywords:** colored wheat, purple-wheat-baked products, anthocyanins stability, health benefits of purple wheat foods

## Abstract

Colored wheats such as black, blue, or purple wheat are receiving a great interest as healthy food ingredients due to their potential health-enhancing attributes. Purple wheat is an anthocyanin-pigmented grain that holds huge potential in food applications since wheat is the preferred source of energy and protein in human diet. Purple wheat is currently processed into a variety of foods with potent antioxidant properties, which have been demonstrated by in vitro studies. However, the health impacts of purple wheat foods in humans still require further investigations. Meanwhile, anthocyanins are vulnerable molecules that require special stabilization treatments during food preparation and processing. A number of stabilization methods such as co-pigmentation, self-association, encapsulation, metal binding, and adjusting processing conditions have been suggested as a means to diminish the loss of anthocyanins in processed foods and dietary supplements. The present review was intended to provide insights about purple wheat food product development and its roles in human health. In addition, methods for stabilizing anthocyanins during processing were briefly discussed.

## 1. Introduction

Colored wheats such as black, blue, or purple wheat are receiving a great interest by the food industry, researchers, and consumers due to their potential health-enhancing attributes. The kernel outer layers of colored wheat contain anthocyanin pigments along with other polyphenols, which are responsible for their health benefits. The pigments in the grain possess different anthocyanin profiles, e.g., blue versus purple [1,2], and a wide range of anthocyanin concentrations that are subjected to genotype, phenotype, and environment interactions [3,4]. The predominant anthocyanin pigments in blue wheat are delphinidin-3-glucoside, delphinidin-3-galactoside, delphinidin-3-rutinoside, and malvidin-3-glucoside [1,5,6]. In purple wheat, cyanidin-3-glucoside, cyanidin-3-(6-malonyl glucoside), cyanidin-3-rutinoside, peonidin-3-glucoside, and peonidin-3-(6-malonylglucoside) are the major anthocyanins [1,7,8,9]. Compared to purple or blue wheat, black wheat has received less attention, perhaps due to the availability of only few genotypes. The main anthocyanin compounds in black wheat are derivatives of the six common anthocyanidins (delphinidin, cyanidin, pelargonidin, peonidin, petunidin, and malvidin (Figure 1)) with glucose and rutinose sugar moieties [10]. On the other hand, the red, common bread wheat, has very little or no anthocyanin pigments [8]. The total anthocyanin content varies widely among colored wheats being 95–277, 22–278, 72–211, and 7–10 µg/g in black, purple, blue, and red wheat, respectively [11]. In Canada, several purple wheat cultivars such as CDC Prime-Purple and CDC Ultra-Prime Purple with exceptionally high contents of anthocyanins (up to 400 µg/g) were developed and are now commercially processed.

In general, consumption of anthocyanins has been associated with lowering the risk of chronic diseases and health promotion [12,13,14,15,16,17,18]. Anthocyanins are important components of the human diet with a daily intake of 12.5 mg in the United States, which is mainly delivered from eating fruits and vegetables [19]. The study has also shown that cyanidin, delphinidin, and malvidin derivatives are the most consumed anthocyanins, accounting for about 45, 21, and 15% of the total anthocyanin intake, respectively. The consumption of anthocyanins varies among countries and regions subject to the type of diet and gender. In Australia, the average daily intake is 24.2 mg [20], and it is 19.8–64.9 mg for men and 18.4–44.1 mg for women in Europe [15]. Nevertheless, anthocyanins are vulnerable molecules and require special stabilization treatments during food preparation and processing. Several review articles have discussed a number of approaches for enhancing the stability of anthocyanin, primarily in fruits and vegetables, including co-pigmentation, self-association, encapsulation and metal binding [21], extraction with advanced extractors [22], interactions with food proteins and polysaccharides [23], or adjustments of processing conditions [24]. In addition, several reports have unveiled the composition and potential health impacts of anthocyanins from cereal grains [13,25] and colored wheats [11,13,26,27,28,29]. The majority of these reviews have focused mainly on the chemistry and genetic attributes. The present review was intended to focus on the development of purple wheat food products and their roles in human health. It presents recent studies conducted on purple wheat in terms of processing and health effects roughly over the past two decades. In addition, methods for stabilizing anthocyanin during processing were discussed. It is the first review that highlights purple wheat food development and its potential in improving human health.

## 2. Purple Wheat Products

Colored wheat foods and products have recently emerged as a true promise to improve human health due to their contents of bioactive components, especially anthocyanins, carotenoids, flavonoids, and phenolic acids. In particular, developing a variety of novel nutrient-dense and health-enhancing staple foods from colored wheat such as breads, pastas, breakfast cereals, and convenience bars would boost healthy eating among general populations and help with reducing the risk factors of chronic diseases. Different food products that have been developed from purple wheat milling fractions are shown in Figure 2. In Canada, several purple wheat products including wholegrain flour, bran, flakes, and others are commercially available under the brand “Anthograin”.

### 2.1. Bread

Bread is the most widely consumed bakery product worldwide providing calories, proteins, fibers, vitamins, minerals, and antioxidants in the human diet. Processing purple wheat into bread products would provide superior nutritional quality over regular wheat bread. For instance, bread made from purple wheat exhibits better nutritional properties due to its content of phenolic acids and anthocyanins and its antioxidant properties compared with regular wholegrain or white wheat bread [30]. In another study, bread made from black wheat has shown a better quality due to its protein characteristics compared with purple or blue wheat bread [31]. Currently, colored wheats are being produced for various food applications, and, undoubtedly, research is needed to breed colored wheat for a specific end use and to guarantee product quality. Chapatti is a non-leavened flat bread commonly prepared from whole wheat and widely consumed in India, Pakistan, and Bangladesh. Chapatti made from black, blue, and purple wheat flours has been found to have higher amino acid retention compared with white wheat chapatti [32]. The average reduction in amino acids content was minimal in chapatti made from black wheat (11.4%) followed by 12.4, 19.0, and 23.8% reductions in the cases of blue, purple, and white wheat chapattis, respectively. The study suggests that anthocyanins in colored wheats may mask and protect proteins and amino acids against thermal and oxidative damages during chapatti production. The sensory properties of colored wheat breads seem exceptional due to the presence of anthocyanins. Chapatti prepared from black, blue, and purple wheat has nicer flavors, softer textures, acceptable quality, and higher contents of dietary fiber and phenolic compounds than white wheat chapatti [32,33]. In spite of reduction during processing, anthocyanin content and antioxidant capacity were highest in black wheat chapatti, followed by blue, purple, and white wheat chapattis. In a study on purple wheat, different milling fractions were prepared and baked into breads to investigate their color properties [34]. Breads made from purple wheat wholegrain flour had a reddish crumb color, while those prepared from the inner milling fractions of purple wheat exhibited a yellowish crumb color.

Anthocyanin compounds are susceptible to heat and oxygen, thus baking affects their composition and content. Breads made from blue wheat wholegrain flour using a traditional Czech recipe baked at 240 °C for 21 min have shown 7.1% reductions in anthocyanin content in comparison with their original flour, whereas a greater reduction (61%) has been found in case of purple wheat bread under the same baking conditions [35]. Breads baked for longer time at lower temperature (e.g., 31 min at 180 °C) have exhibited even greater reductions of 40.8 and 72.8% in anthocyanin content in blue and purple wheat, respectively. Studies have indicated that longer baking time, rather than baking temperature, had a greater impact on the anthocyanin content of bread. Two prototypes of purple wheat breads, wholegrain bread and bran-enriched wholegrain bread, were developed as part of several purple wheat products to investigate their anthocyanin composition and antioxidant activity [36]. The bran-enriched bread had about 17 µg/g anthocyanin content, while the wholegrain purple wheat bread had 10 µg/g, as measured by HPLC. Higher values of anthocyanin content, 80 and 65 µg/g, were observed in both type of bread when the measurement was conducted by a spectrophotometric method [36]. These results indicated significant reductions of anthocyanin during bread production compared with the original flours. Similar observations have been reported for the concentration of anthocyanins in breads made from blue and purple wheat [37]. The majority of the anthocyanin loss in blue wheat occurs during baking (45–51%), whereas the biggest loss in purple wheat takes place during dough preparation (26–39%). When making bread, the blue wheat loses an average of 75–77% of its anthocyanins, whereas the purple wheat loses an average of only 50–53%. Among the individual anthocyanin compounds, peonidin-, cyanidin- and petunidin-glycosides are the most stable during bread production, while delphinidin-glycoside is the lowest stable anthocyanin [37]. Peonidin-3-glucoside has a loss percentage of 44%, cyanidin-3-glucoside of 52%, cyanidin-3-rutinoside of 61%, and delphinidin-3-rutinoside of 80%. Another study looked into how the bread making process affected the phenolic content and antioxidant qualities of purple wheat [38]. The free phenolic content significantly increased (*p* < 0.05) after mixing, fermentation, and baking, while no significant changes have been found in bound phenolics after fermentation for 30 min. However, a longer fermentation time (65 min) and baking for 25 min at 200 °C resulted in a significant increase (*p* < 0.05) in bound phenolic compounds by 16% to 27%. A similar trend has been observed for the antioxidant properties. After dough mixing, total anthocyanin content (TAC) significantly decreased (*p* < 0.05) by 21%, then it gradually increased to 90% of the original level after fermentation. Baking resulted in a significant reduction (*p* < 0.05) in TAC by 55%. The overall loss of TAC is greater in bread crust than that of bread crumb. Similarly, the anthocyanin concentration dramatically dropped after mixing and baking of purple and blue wheat bread doughs, but not after the fermentation or proofing steps [39]. Although the anthocyanin loss during baking cannot be completely prevented, it can be slowed down or controlled by altering the variables of the bread-making process. This may include using high-temperature, short-time baking processes, as well as a dough making process at a low pH [39,40]. For example, the fermentation of purple wheat with *Lactiplantibacillus plantarum* No. 35 resulted in improved DPPH free radical scavenging capacity, and it has been recommended for acrylamide reduction in bread [41]. Thus, the application of the sourdough processing technique could be an effective method to protect anthocyanins and other phenolic compounds during the production of fermented baked products from colored wheats. Other bran-enriched products with no fermentation steps and minimal baking conditions (shorter time and lower temperature), such as crackers, may have higher anthocyanin content and lower reduction percent than breads. In general, the baking process could induce positive and/or negative effects on product quality and nutritional properties of breads made from colored wheat subject to bread type, fermentation method, and baking temperature and time.

Purple wheat milling fractions were used to create ready-to-bake flour mixtures suitable for bread production [42]. The mixtures contained wholegrain purple wheat flour and/or purple wheat white flour enriched with fiber sources such as inulin, chia seed flour, and psyllium husk flour to increase their fiber content and biological value. The purple wheat white flour mixture contained 263 µg/100 g total phenols and produced bread with a light color, acceptable texture, taste, and flavor, while the mixture made from 70% wholegrain and 30% white flour had a higher level of total phenols (897 µg/100 g) and produced bread with enhanced consistency, flavor, and aroma. The overall quality of both breads was acceptable. Purple and blue wheat flours were successfully used in the production of buns, a type of bread roll used mainly in making sandwiches [43]. The doughs and buns were prepared with wholegrain flour from regular wheat, purple wheat, or blue wheat, either by itself (100%) or in combination with white flour (90/10 and 80/20). Purple wheat flour had the maximum dough stiffness, while blue wheat exhibited the lowest dough stiffness among flours. Buns made from blue wheat had significantly lower crumb hardness (*p* ˂ 0.01) than control buns made from ordinary wheat, which had the maximum crumb hardness. The study results suggested that the buns made from reduced levels of colored wheat flour were springier. Overall, these studies suggested that colored wheat, including purple wheat, is a promising flour ingredient in making different types of breads, but it requires special preparation and baking conditions to minimize the loss of anthocyanins.

### 2.2. Pasta and Noodle

Pasta is a popular food around the globe, which is produced either in fresh or dry form. It is considered a low glycemic index food that elicits low postprandial blood glucose and insulin responses [44]. The inclusion of colored wheat ingredients in pasta recipes would add further nutritional value to pasta products. At a pilot plant scale, a fiber-rich fraction obtained from purple wheat through the debranning process was added to semolina or flour formulations at a 25% level to produce nutritious pasta [45]. The fiber-enriched pasta products were found to contain reasonable amounts of total anthocyanin (89–122 µg/g) and total phenols (3000–3100 µg/g). Cooking of pasta dropped the total anthocyanin and phenol contents significantly, by about 58–65% and 51–60%, respectively. The HPLC data has shown that the abundant glycosylated anthocyanin moieties are more likely to be released during cooking than their aglycones. The anthocyanin aglycones could be retained by the pasta dough matrix, regardless of whether the enriched pasta was based on semolina or wholegrain flour. In another study, purple wheat was debranned into two fractions, fraction 1 (F1, 3.7% of outer layers was removed) and fraction 2 (F2, additional 6% of outer layers was removed), and compared with conventional milling bran (CB) in making fiber-enriched pasta [46]. Pasta enriched either with F1 or F2 had significantly higher amounts of anthocyanins than those enriched with the CB bran. Additionally, pasta enriched with F1 had the highest antioxidant capacity. In another study, purple durum and non-anthocyanin-pigmented durum wheats were milled by roller or stone mill, and the milled products were compared in the pasta making process [47]. Stone milling has been found to preserve health-enhancing compounds such as dietary fiber, carotenoids, and anthocyanins (for purple durum wheat) than roller milling. During the pasta making process, the total anthocyanin content showed a gradual decline from 66.5 µg/g in the raw material to 49.3 µg/g after extrusion. The drying process resulted in significant (*p* < 0.05) reductions in the content of anthocyanins (21.4 µg/g vs. 46.3 µg/g) and carotenoids (3.77 µg/g vs. 4.04 µg/g), but slight changes in antioxidant capacity have been observed. It has been suggested that some modifications in the processing conditions such as moisture content and drying temperature could be made to preserve more anthocyanins and carotenoids. Pasta made from either pigmented or ancient wholegrain wheats have shown acceptable quality due to low cooking losses and comparable physical characteristics to semolina pasta [48]. In comparison to durum and ancient wheat semolina, pasta made with pigmented wheat have much higher total phenolic contents and antioxidant activities [48]. Despite the reduction in anthocyanins, the remaining portion of anthocyanins in purple durum pasta could be beneficial to human health.

Noodles are a staple food in several parts of the world. The color of noodles is a key quality characteristic and has a significant role in influencing consumer acceptance. Purple wheat milling fractions were used in making noodles to study their impact on product appearance [34]. The color of noodles made from wholegrain purple wheat flour or a combination of 10% bran and middle fraction exhibits a strong red hue. The addition of black, blue, purple, and white wheat bran powders, prepared by ultrafine grinding into Chinese noodles formula at levels of 2–6%, has resulted in better texture and lower cooking loss compared with the control noodles with no fiber added [49]. The addition of bran powder reduced the degree of whiteness (L* values) of wet dough sheets, which is mostly due to the presence of carotenoid and anthocyanin pigments. The study highlighted the possibility of producing fiber-enriched noodles with potential antioxidant capacity by using wheat bran powder with different colors. Another study examined the phenolic antioxidant properties of noodles prepared from whole wheat, partially debranned grain, and refined flours of three colored wheats (dark purple, light purple, and black) [50]. The total phenol and flavonoid contents and antioxidant capacity of the noodles were lower compared to their original flours. Noodles prepared from a blend of regular and black or purple wheat flours have been found to possess higher amounts of total phenols and anthocyanins than that made from regular wheat flour alone [51]. These studies show that adding colored wheat milling fractions to pastas and noodles improve their content of anthocyanins, polyphenols, and antioxidant properties.

### 2.3. Other Cereal Products

Purple wheat flour has been considered in making biscuits and crackers with the intention to improve their nutritional properties and supply the market with healthy food products. Anthocyanin-enriched biscuits made from wholegrain purple wheat flour were developed [52]. The product contains 2.6 mg/g total phenol content and 13.9 µg/g total anthocyanins, which indicates an improved nutritional profile compared with the control biscuits. Due to their superior antioxidant qualities, the purple wheat biscuits had lower levels of lipid-derived carboxylic acids, indicating a slower rate of oxidative degradation of lipids. On the other hand, the purple wheat biscuits showed lower texture quality than the conventional biscuits, perhaps due to the higher gluten index of the purple wheat flour, which impacts biscuits dough rheological properties.

Crackers made from wholegrain purple and blue wheat flours have been found to contain 37–45 µg/g anthocyanins [39]. This level is about 70% lower than that of the initial flour. The majority of the anthocyanin content of purple wheat flour (50%) is lost during dough mixing, whereas blue wheat flour loses more anthocyanin during baking than during mixing. This might be explained by the location of anthocyanin pigments in both types of wheat (e.g., in the pericarp of purple wheat and in the aleurone layer of blue wheat) and their interactions with other components during mixing, where oxidative stress damage takes place, while heat damage occurs during oven baking [39]. A bran-enriched high fiber cracker with a high amount of anthocyanins (56 µg/g) was developed as a functional food [36]. In spite of this high level, a significant decline in anthocyanin content occurred through the processing, with more than 80% reduction compared with the starting material (blend of whole flour and bran at 1:1 ratio). The crackers had acceptable sensory properties and exhibited high antioxidant activity, as demonstrated by in vitro assays based on ABTS, DPPH, and peroxyl radical activity. Four servings of the crackers provided 6.7 mg anthocyanin and 176 mg phenolic acids and have the potential to maintain positive health impacts.

Several purple wheat products (Figure 2), including bran-enriched convenience bars, crackers, pancakes, and porridge, were also developed [36]. These products were developed with the intention to study their potential impact on metabolic markers and health conditions, as described in Section 4. The products were significantly different in their nutrient level, in particular their anthocyanins and dietary fibers. The purple wheat bran was incorporated to increase the amount of dietary fiber and anthocyanins in the bars. As a result, the bars had a great anthocyanins concentration (41.7 µg/g) compared to 16.0 and 7.0 µg/g for pancake and porridge.

Purple wheat bran-enriched muffins were developed as healthy foods to assess the impact of thermal processing on their antioxidant properties [53]. The muffin-making process had significant adverse effects on the phenolic compounds of wheat, especially anthocyanins. In spite of the complete decay of anthocyanins, the purple wheat muffins exhibited good DPPH scavenging activity after thermal processing. Home- and laboratory-made infant cereals prepared from whole purple wheat, unpolished red rice, and partially polished red rice were evaluated in terms of their total phenol and anthocyanin contents and total antioxidant capacity [54]. Infant cereals made from colored grains, purple wheat, or red rice have a higher phenol content and greater ORAC values than commercial infant cereals (*p* ˂ 0.05). Moreover, the unpolished red rice cereals have a total phenol content and peroxyl radical scavenging capacity higher than those made from purple wheat, while the latter has a higher total antioxidant capacity, suggesting that giving infants this grain in their diets may improve their antioxidant status as well as the overall body wellness.

### 2.4. Anthocyanin-Rich Powder

Purple wheat can be a sustainable source of anthocyanin pigments, as they are concentrated in the outer layers of the kernel. A mechanical–chemical process was developed to isolate anthocyanins from purple wheat [7]. Firstly, the grains were milled and fractionated to obtain the bran fraction with a 2-fold increase in anthocyanin concentration, then the anthocyanins were extracted from the bran with acidified ethanol. The extract was concentrated by evaporation at 40 °C using a rotary evaporator and purified on a chromatographic column filled with amberlite XAD-7HP packing material. The ethanol elution from the column was concentrated again in a rotary evaporator and then dried in a solvent proof oven at 45 °C to obtain the anthocyanin-rich powder, with an 81- to 135-fold increase in anthocyanin concentration, subject to batch size. The powder had an exceptionally high content of anthocyanins (3.4–5.7 g/100 g), with 7.9–9.9% moisture content. Cyanidin was the main aglycone, along with glucose as the dominant sugar, and malonyl being the main acyl substituent in the acylated pigments present in the purple wheat bran or powder. Peonidin came second after cyanidin, and the other common aglycones (delphinidin, petunidin, pelargonidin and malvidin) were also present, but at very low concentrations. The purple wheat bran and powder products exhibited potent antioxidant capacities based on the scavenging of DPPH, ABTS, and peroxyl radicals compared with wholegrain flour. The study suggested that processing of purple wheat into bran and anthocyanin-rich powder would add value to the bran and expand its use as a renewable source of anthocyanin pigments for the functional foods, nutraceuticals, cosmetics, and healthcare industries. The same process was previously used to isolate anthocyanins from blue wheat, which was further fractionated into individual anthocyanin components using preparative HPLC equipped with an analytical fraction collector [55]. These studies indicated that it is feasible to covert colored wheat bran into value-added products.

## 3. Improving Stability of Anthocyanins

Stability of anthocyanins is a crucial issue in the development of anthocyanin-rich foods and natural health products including purple wheat foods [36] and purple wheat anthocyanin-rich powder [7] due to their vulnerability to processing conditions. A number of intrinsic and extrinsic variables could be considered for the enhancement of anthocyanin stability in processed foods and, eventually, their anticipated beneficial health effects and color properties. These variables could be moderated during processing of anthocyanin-pigmented grains such as purple wheat to protect anthocyanin compounds against thermal and oxidative damage. Structures of common anthocyanidins depicting their non-acylated and acylated forms and potential stabilization reactions are given in Figure 1. The first stabilization approach involves altering the structure of anthocyanin compounds via inducing intermolecular interactions with other food components or ingredients during processing, such as non-colored flavonoids, proteins, or metals. Several structurally modified interactions could be induced during processing or storage of anthocyanin-rich foods and beverages in order to maintain the intensity of color and bioactivity of anthocyanins. Examples of these methods include intra- and inter-copigmentation, glycosylation, acylation, and methylation reactions. The development of colored wheat varieties having more stable anthocyanin profiles (e.g., acylated and/or di-glycosylated) that could withstand processing conditions might also be taken into consideration under this approach. A second approach is to control processing variables in order to diminish thermal and chemical damages to anthocyanins. A third way to protect anthocyanins from degradation is encapsulation [21]. Combinations of the above methods could be possible; however, cost effectiveness and feasibility should be of prime importance if the method is considered for implementation at the industry level.

Copigmentation is the most commonly used method for anthocyanin stabilization. It is a non-covalent interaction between anthocyanin (flavylium cation or quinonoid base) and a colorless phenolic compound, referred to as a co-pigment, such as phenolic acids and flavonoids, which results in a hyperchromic shift and superior pigment stability. To the best of our knowledge, nothing has been reported on the copigmentation of purple wheat anthocyanins, but several reports have been published on the copigmentation of anthocyanins in wine and fruits [56,57,58,59]. The copigmentation reaction may occur during the processing and storage of foods or beverages at low pH, with the most effective pH value being 3.3 for grape skin extract [60]. Thus, it is preferred to incorporate anthocyanin ingredients such as purple wheat bran, flakes, or anthocyanin powder in acidic food formulations such as sour dough and yoghurt products to enhance their appearance, shelf life, and health benefits. For instance, it has been reported that adding purple corn anthocyanins to light-protected milk prevents lipid oxidation and improves antioxidant and sensory properties [61].

Methylation is the increasing of the number of methoxyl groups in anthocyanidins (aglycones), resulting in lowering hydroxyl groups and improving anthocyanin stability. Among the six common anthocyaninidins (Figure 1), malvidin is the most stable, followed by peonidin, petunidin, pelargonidin, cyanidin, and delphinidin [56]. A study has shown that malvidin-3-galactoside and malvidin-3-arabinoside have resistances to alkalis and heat, followed by petunidin-3-galactoside [62]. Due to variations in gene expression patterns, the methylation mechanism could also be applied in breeding programs aimed at the development of a highly stable anthocyanin profile in anthocyanin-pigmented plant materials [63]. Metal complexation or chelation of anthocyanin pigments could also play a significant role in the stability of anthocyanins, especially in fruit juices and wine, through complexing with free hydroxyl groups in anthocyanidins resulting in bathochromic and hyperchromic shifts [21]. When trivalent ions, such as iron, are used instead of divalent ions, anthocyanins derived from vegetable and fruit extracts undergo a greater bathochromic shift [64]. This mechanism may hold promise for juices and yoghurts that contain anthocyanin-rich plant materials such as purple wheat grits or flakes as natural colorants and/or antioxidants.

Glycosylation and acylation of sugars also enhances stability of anthocyanins. They are essential reactions in anthocyanin biosynthesis and could be implemented to improve stability of anthocyanins. The higher the number of glycosyl groups, the higher the stability of anthocyanins, i.e., diglycosides are more stable than their corresponding monoglycoside. Details about glycosylation and its enhancing effects on the stability of anthocyanins have been previously reported [65]. The structure of anthocyanins can also be stabilized by interactions with other food components such as proteins and polysaccharides [66,67,68]. In general, the molecular interactions of purple wheat anthocyanins with proteins or other components in food systems have not been studied. Since purple wheat has the potential as a functional food ingredient, this area warrants investigation to determine how interactions with food components can improve the stability, bioavailability, and bioactivity of anthocyanins. The availability of anthocyanin-enriched foods with improved nutritional and functional attributes and longer shelf lives would boost the intake of anthocyanins and antioxidants.

Encapsulation is a technology employed to protect bioactive compounds by entrapping them inside solid particles or liquid vesicles to help stabilize them and control their release. It improves the stability and delivery of light-, acid-, and heat-labile molecules such as anthocyanins. Due to its relatively high cost, this technology would be suitable for the development of nutraceutical and dietary supplement products but inappropriate for food applications. It has been reported that anthocyanins encapsulated in emulsions appeared to reduce pH-induced color changes, perhaps due to the difference in pH between the inner water phase and the outer water phase [69]. Various methods, including freeze-drying, spray drying, supercritical carbon dioxide, and others, have been employed in encapsulation to create a variety of systems with different sizes, structures, shapes, surface characteristics, stability, and carrier materials [70]. The developed purple wheat anthocyanin-rich powder [7] could be a promising functional ingredient for making nutraceutical-, cosmetic-, and healthcare-encapsulated products.

Anthocyanins experience a number of adverse changes during the thermal processing of purple wheat, such as baking and kernel flaking, including thermal and chemical degradation, oxidation, and isomerization [24,36]. These changes mostly induce the cleavage of the pyrylium ring, resulting in structure interruption and molecular rearrangement [71]. These undergo rearrangements that could disturb the stability of anthocyanins and, subsequently, their color and bioactivity [71,72]. By adjusting the processing conditions and/or inducing synergistic interactions with anthocyanins through formulations, these undesirable changes to anthocyanins could be reduced. The stability of anthocyanins is affected by pH, temperature, oxygen, light, presence of enzymes, metals, flavonoids, phenolic acids, and others [73]. Anthocyanins are more stable in the acidic media (low pH) due to the formation of flavylium cation, the most stable cation of anthocyanin conjugates [73]. Other forms of anthocyanin such as colorless pseudobases and hydrolases, which are present in mild acidic or non-acidic media, are unstable, and this influences the color of processed foods [73]. In general, it is important to optimize processing variables and formulations of anthocyanin-rich foods such as purple wheat to avoid and/or lower the degradation and oxidation of anthocyanins and to preserve color-imparting and health-enhancing properties.

## 4. Health-Enhancing Properties of Purple Wheat Foods

Purple wheat is gaining popularity in the food market due to its potential health benefits. The availability of additional purple wheat food choices, particularly staple foods, would boost the consumption of anthocyanin and other phenolic antioxidants. Regular daily consumption of cereal and baked goods, which are staples in the diets of many populations, may offer a practical and cost-effective solution in the battle against many health-related problems such as obesity, diabetes, hypertension, and inflammation, which would improve the overall health advantages.

### 4.1. Antioxidant Properties

Purple wheat grains and foods possess high antioxidant capacity largely linked to their content of anthocyanins and other phenolic compounds. These bioactive compounds are mostly present in the outer layers of kernels, which make purple wheat wholegrain flour and bran healthy food ingredients. These polyphenolic compounds offer several health benefits, including potent antioxidant and anti-inflammatory properties [13,26,74]. In general, consumption of wholegrain foods is associated with reduced risk of inflammation, heart diseases, several forms of cancer, and improved blood glucose and insulin resistance in human and animal models [75,76,77]. The antioxidant and anti-inflammatory properties of anthocyanins and phenolic acids are dependent on their concentrations and the molecular structures of grain foods, subject to bioavailability. It is essential to maintain a low level of oxidative stress for the normal functioning of cell-signaling pathways and communication in the human body. Complications occur when free radicals build up in the body, causing damage to cells and cell components, such as DNA and proteins, which trigger inflammation and the development of chronic diseases [78].

Purple wheat offers a food ingredient with better antioxidant properties and the ability to scavenge DPPH (1,1-diphenyl-2-picrylhydrazyl) radicals at a higher level (7.1–8.6 µM Trolox equivalent/g) than that of yellow or red wheat (6.5–6.7 µM Trolox equivalent/g) [79]. The DPPH scavenging capacities of the free phenolic extracts of 40 purple wheat genotypes have been found to exhibit a range of 40–60%, compared with 43–63% for the bound phenolic fraction [80]. In fact, the consumption of purple wheat, having a total anthocyanin content of 41.7 mg/Kg, by experimental animal groups (rats *n* = 32, chickens *n* = 32 and fish *n* = 20) has shown a significant positive impact on antioxidant activity and function of the liver tissue in rats and chickens in comparison to a control group fed regularly with wheat with a 25.0 mg/kg anthocyanin content [81]. The antioxidant activity was based on DPPH, FRAP (Ferric Reducing Antioxidant Power method), and ABTS (2,2′-azino-bis(3-ethylbenzothiazoline-6-sulphonic acid)), while the liver function was based on an increase in blood enzymatic activities such as γ-glutamyl transferase, aspartate aminotransferase, alanine aminotransferase, and alkaline phosphatase. The fact that there were no significant differences in the hepatopancreatic and blood enzymes between the two groups of fish fed purple wheat versus regular wheat diet suggests that fish cannot metabolize anthocyanins.

Purple wheat milling fractions differ in their content of anthocyanins, phenolics, and antioxidants, offering an opportunity to make a variety of food ingredients vary in their technological and nutritional functionalities. Among the milling fractions, purple wheat bran has the highest level of anthocyanins, followed by wholegrain flour, while white flour has the lowest concentration [8]. Purple wheat flour, bran, and flakes were used to prepare various food products including bread, porridge, pancakes, crackers, and convenience bars, which exhibit a wide range of anthocyanin compositions and antioxidant capacities [36]. Among the products, the crackers and convenience bars have shown great potential to inhibit radical activity at 76% and 80% inhibition activity of DPPH and ORAC, respectively, indicating their potential to impact oxidative stress in humans. It has also been reported that purple wheat bran-enriched muffins exhibit reasonable levels of anthocyanins along with promising antioxidant properties, despite the significant reduction in total phenolic content and ORAC values during the processing of muffins [53]. Similarly, ABTS values have been found to increase significantly by replacing portions of common wheat flour with purple wheat bran in fresh noodles, e.g., 79 µmol TE/100 g for regular wheat noodles compared with 246, 315, and 487 µmol TE/100 g for noodles prepared with 10, 30, and 50% replacement levels of purple wheat bran [51]. A similar trend has been observed in the case of DPPH with a 3- to 4-fold increase in noodles containing purple wheat bran. Home- and laboratory-made infant cereals made from purple wheat have been reported to possess higher cellular antioxidant than those made from partially polished or unpolished red rice, indicating the potential of purple wheat in making infant cereals [54]. These studies have shown that the consumption of purple wheat foods would provide reasonable levels of bioactive compounds and antioxidants in a diet that boosts their daily intake. A summary of studies showing in vitro antioxidant capacity of purple wheat products is given in Table 1.

### 4.2. Health Benefits

The health benefits of colored grain foods have become well documented, based on in vitro and in vivo human and animal studies. Although many in vitro studies have been conducted on purple wheat, with a particular interest in assessing their potential antioxidant and health effects, only few human studies are available. A summary of animal and human studies on the health impacts of purple wheat diet is presented in Table 1.

#### 4.2.1. Animal Studies

In a nutrigenomics study, mice were fed a high fat diet supplemented with white, purple, or black whole wheat for 12 weeks. The purple and black wheat diets contained 40 and 150 µg/g total anthocyanin contents, respectively [6]. Both colored wheats lowered serum concentrations of total cholesterol, triglyceride, and free fatty acids, and restored blood glucose levels and deceased insulin resistances. The black wheat diet significantly induced an increase in the expression of enzymes involved in fatty acid balancing, β-oxidation, and oxidative stress. Additionally, transcriptome analysis of adipose and liver tissues showed an activation of numerous pathways and genes related to fatty acid-oxidation, anti-oxidative enzymes, as well as balancing fatty acid metabolism, specifically in the black-wheat-supplemented mice group. In a different study, consuming a purple wheat diet for six weeks has been found to reduce lipid metabolism disorder and liver tissue as well as renal injury in hyperlipidemic rats [82]. A total of 42 rats were randomly divided into four groups, with the normal control group receiving a standard diet, and the other three groups being fed a high-fat diet to induce dyslipidemia. Two groups of the dyslipidemic rats were fed with a diet containing 60% purple wheat flour derived from two different cultivars, while the dyslipidaemia control group was fed 60% regular wheat. The triglyceride, total cholesterol, and low-density lipoprotein levels dramatically decreased to nearly normal levels in the two groups given purple wheat diets. Additionally, the purple wheat diet reduced degeneration of the fatty liver tissue along with mitigation of lipid metabolism disorder and renal injury, which might reverse kidney damage and hepatocyte steatosis caused by dyslipidemia [82]. On the contrary, feeding broiler rabbits (*n* = 18) a pelleted feed mixture containing 15% purple wheat for 61 days showed no significant impact (*p* > 0.05) on plasma biomarkers, enzyme activity, hepatic metabolism, total bilirubin concentration, triglycerides, cholesterol, urea, globulin and albumin contents, and antioxidant activities in comparison with the control regular wheat group [83]. Interestingly, a study investigating potential antiaging and antioxidant properties of purple wheat using nematode (*Caenorhabditis elegans*) as an experimental organism has shown that the anthocyanin-rich extract of purple wheat extends the mean life span of wild-type worms and mev-1(hn1) mutants, which are sensitive to oxidative stress, by 10.5 and 9.2%, respectively [84]. It has been reported that the purple wheat treatment activates translocation of DAF-16/FOXO, the life span extension inducer, to the nucleus which inhibits the insulin/IGF-1-like signaling pathway. A study investigating the effect of two isogenic wheat lines that differ in their content of anthocyanins on neurodegenerative disorders in mouse models demonstrated the role of anthocyanins to improve cognitive functions [85]. The wheat diets were fed to a mouse model of Alzheimer’s disease, driven by amyloid-beta (Aβ) injection and a transgenic mouse model of Parkinson’s disease (PD) with an overexpression of human alpha-synuclein for 5–6 months. Wheat anthocyanins have shown the ability to prevent deficits in working memory induced by Aβ and partially restore changes to episodic memory. In the PD model, both wheat diets prevented memory loss and restored its facilitation. It has been suggested that anthocyanin-rich wheat is a promising diet component to support the early stage of neurodegenerative disorders.

#### 4.2.2. Cell Culture Studies

Extracts from purple, blue, and black wheat have been reported to reduce nitrite oxide production in the lipopolysaccharide-induced pro-inflammatory stress cellular model (murine macrophage cell line) and to exhibit anti-inflammatory effects via the inhibition of the pro-inflammatory cytokines’ (TNF-α and IL-1β) production [86]. In another study, human intestinal epithelial Caco-2 cells were employed as an in vitro model to study the anti-inflammatory properties of pasta made from wheat flour or semolina containing 25% purple wheat bran [45]. The serving amount of dry-weight-enriched pasta (60 g) provided up to 2.4 mg anthocyanin after cooking. The estimated anthocyanin amount released in the duodenum following digestion was around 4–6 mg/L. This amount was able to partially inhibit the activity of enzymes involved in glucose metabolism, thus reducing glycemic responses. The ethanol extract of anthocyanin-rich bran pasta suppressed the IL1β-stimulated expression of NF-кB in the cellular model, which demonstrated its potential anti-inflammatory effects. In general, anthocyanins have been shown to trigger several therapeutic effects against inflammation, cardiovascular diseases, hyperglycemia, and oxidative liver damage [13,28,87].

#### 4.2.3. Human Studies

To the best of our knowledge, only two human studies were conducted to investigate the bioavailability and health impacts of anthocyanins and phenolic acids in individuals who consumed purple wheat crackers and convenience bars [88,89]. In the first acute consumption study, changes in plasma antioxidant status and short-term markers of inflammation were investigated following the consumption of four servings of bran-enriched purple wheat convenience bars or crackers containing about 6.7 mg anthocyanins and 176–213 mg total phenolic acids by healthy individuals. Only a few phenolic acids and their metabolites have been detected in the plasma, but no intact anthocyanin compounds have been found [88]. In addition, low levels of anthocyanin and phenolic acid metabolites have been detected in the subjects’ urine, which may suggest that the products were extensively digested, metabolized, and used by the human body. Nevertheless, no significant increases in IL-6 and TNF-α inflammatory indicators or plasma antioxidant activity have been noted due to short-term consumption (8 h). The second study examined the concept of whether daily consumption of wheat products could result in the accumulation of phenolic compounds in the body and further induce positive metabolic and health impacts [89]. In the latter study, a randomized, single-blind, parallel arms human clinical trial was conducted to determine the effect of consuming purple wheat anthocyanin-rich convenience bars for eight weeks by overweight and obese, non-smoker adults, with stable medications and mild inflammation (hs-CRP > 1 mg/L), on oxidative stress and inflammatory responses compared to regular wheat wholegrain bars. The regular wheat wholegrain convenience bars contained 190–215 mg phenolic acids and 40–44 g dietary fiber per 4 servings (total of 160 g), and the purple wheat bars had an additional 1.7 mg of anthocyanins. A significant reduction in IL-6 and an increase in adiponectin was observed within the purple wheat group. In general, the consumption of purple or regular wheat products has shown the ability to enhance plasma markers of inflammation and oxidative stress in participants with evidence of chronic inflammation, with minor variations, depending on the type of wheat [89]. Comparing the two kinds of wheats, only plasma TNF-α and glucose have been found to be significantly (*p* < 0.05) different, i.e., TNF-α and glucose decreased in regular and purple wheat groups, respectively. This showed the role of purple wheat products in the management of obesity and diabetes, which is mainly related to its phenolic content, especially anthocyanins. The anti-diabetic effect of anthocyanins is linked to several mechanisms, which include the inhibition of carbohydrate digestive enzymes, modulation and/or supersession of important transcription factors in the metabolism of carbohydrates such as adenosine monophosphate-activated protein kinase (AMPK), proliferator-activated receptor gamma (PPARγ) and nuclear factor κB (NF- κB), and activation of phosphoinositide 3 kinase/protein kinase B (PI3K/AKT)-mediated energy metabolism [90]. Further human studies are needed to validate the physiological effects of purple wheat products, which would support the promotion of purple wheat foods with proven positive health impacts.

**Table 1 foods-12-01358-t001:** Summary of in vitro and in vivo studies conducted on purple wheat products.

Wheat Type	Food Type	Model Used	Subjects and Assay	Health Effects	References
Purple, black, and white wheat	High fat diet supplemented with wheat	Animal study (12 weeks)	Male Swiss albino mice (age 6–7 weeks, 20–22 g)	Both black and purple wheats reduce total cholesterol, triglyceride, and free fatty acid levels in serum, with the restoration of blood glucose and insulin resistance	Sharma et al., 2020 [6]
Purple wheat	Bran and anthocyanin-rich powder	In vitro study	ABTS, DPPH, and ORAC assays	Exceptional antioxidant properties	Abdel-Aal et al., 2018 [7]
Purple wheat	Bran-enriched crackers and convenience bars	In vitro study	ABTS, DPPH, and ORAC assays	Exceptional antioxidant properties	Gamel et al., 2019 [36]
Purple and yellow wheat	Bread	In vitro study	ABTS and DPPH assays	Bread (crust and crumb) made from purple wheat has higher antioxidant activities	Yu and Beta, 2015 [38]
Purple wheat and durum semolina	Anthocyanin-rich pasta (25% bran)	Cell culture	Human intestinal epithelial Caco-2 cells	Both types of cooked pasta suppress IL1β-stimulated expression of NF-кB in the cellular model	Parizad et al., 2020 [45]
Purple and common wheat	Fresh noodle	In vitro study	ABTS and DPPH assays	Increased antioxidant capacity with the addition of purple wheat bran	Park et al., 2022 [51]
Purple wheat	Home- and laboratory-made whole purple wheat infant cereals	In vitro cellular antioxidant activity In vitro cellular proliferation	Primary human fetal small intestine cell line (FHs 74 Int, CCL-241, American Type Culture Collection (ATCC), Manassas, VA)	Higher cellular antioxidant activity than lab made red rice and commercial infant cereals No toxicity against the fetal small intestine cell line	Hirawan et al., 2011 [54]
Purple and common wheat	Milled wheat in pelleted form Coarse meal Crushed wheat kernel	Animal study	Wistar Albino male rats (age of 9 weeks, *n* = 64). Chickens of the hybrid combination COBB 500 (age of 39 days, *n* = 32) Fingerlings of common carp (*Cyprinus carpio* L.) (*n* = 100).	Significant higher antioxidant status in the liver of rats and chickens fed purple wheat No significant differences in hepatopancreas enzymes of fish	Mrkvicová et al., 2016 [81]
Purple and regular wheat	Diet containing 60% purple or regular wheat High-fat diet to induce dyslipidaemia	Animal study (6 weeks)	Dyslipidaemic male rats (weight 180–210 g, *n* = 42)	Reduced triglyceride, total cholesterol and low-density lipoprotein, fatty liver, and mitigation of lipid metabolism disorders and renal injury in groups fed purple wheat diet	Lan et al., 2022 [82]
Purple and common wheat	Pelleted feed (purple wheat plus anthocyanin)	Animal study (61 days)	Broiler rabbits (*n* = 18 HYLA female rabbits, age of 32 days)	No significant effects on plasma biomarkers, oxidative stress enzymes, and antioxidant activity	Stastnik et al., 2019 [83]
Purple wheat	Anthocyanin-rich methanol extract	Animal model (Study duration = the whole life span of the Nematodes)	Wild type strain N2 worms of nematode *Caenorhabditis elegans* and mev-1(hn1) mutants	10% extension of life span	Chen et al., 2013 [84]
				Inhibition of insulin/IGF-1-like signaling pathway Increased stress response and reduced oxidative stress	
Purple and common wheat	Wheat diet	Animal study (5–6 months)	Male mice of C57Bl/6J strain (age 2.5 months, Neurodegenerative disorder induced by central injection of an amyloid beta	Prolong memory extinction and improve neurodegenerative disorder	Tikhonova et al., 2020 [85]
Purple, blue, black, and white wheat	Acidified methanol extract	Cell culture	Murine macrophage cell line RAW 264.7	Reduced nitrite oxide production in lipopolysaccharide-induced pro-inflammatory stress.	Sharma et al., 2018 [86]
				Inhibition of pro-inflammatory cytokines (TNF-α and IL-1β)	
Purple wheat	Bran-enriched crackers and convenience bars	Human study— A randomized, semi-blinded crossover acute Study	Healthy participants, 4 servings (6.7 mg anthocyanins and 176–213 mg phenolic acids, plasma antioxidant status and short-term markers of inflammation markers	Few anthocyanin metabolites in urine and none in plasma No short-term impact on plasma antioxidant activity or inflammatory biomarkers, IL-6 and TNF-α	Gamel et al., 2019 [88]
Purple wheat and regular wheat	Bran-enriched convenience bars	Human study A randomised, single-blind parallel-arm study for 8 weeks.	Overweight and obese adults (*n* = 29) with chronic inflammation (high-sensitivity CRP > 1 mg/L)	Significant reduction in IL-6 and increase in adiponectin within the purple wheat group and lower TNF-α in both groups comparing to the starting point	Gamel et al., 2020 [89]

## 5. Conclusions

Since wheat is an important component of the human diet in many parts of the world, the consumption of products made from purple wheat may help individuals boost their daily intake of anthocyanins and phenolic acids and improve the antioxidant status in the body. Purple wheat holds great potential for being processed into a variety of staple foods, such as bread, pasta, and cereal breakfast. However, depending on the food type and processing conditions, the processing of purple wheat results in considerable reductions in anthocyanins. The loss of anthocyanin during processing may not be avoided, but it could be controlled and eased by several approaches. For instance, in making purple wheat breads, the use of the sour dough method along with short-time and low-temperature baking could lessen the loss of anthocyanins. The use of purple wheat bran, grits, or flakes in acidic foods such as yoghurt could be a proper way to protect anthocyanins. It also would improve the quality and nutritional properties of the products by increasing their content of fiber and anthocyanins and enhancing their appearance and textural properties. Several stabilizing mechanisms for anthocyanin could be implemented in purple wheat processing such as copigmentation with non-color flavonoids or phenolic acids and molecular interactions with proteins or polysaccharides in food systems. Since purple wheat has potential as a functional food ingredient, these interactions in food systems warrant investigations to improve the stability and eventual bioavailability and bioactivity of anthocyanins. In general, it is important to prevent and/or reduce anthocyanins’ degradation and oxidation during the processes of anthocyanin-rich foods such as purple wheat. This would increase the stability and, ultimately, the bioavailability and bioactivity of anthocyanins and preserve the color-imparting and health-enhancing properties of the products. The management of obesity, diabetes, oxidative stress, and inflammation with purple wheat food products has been demonstrated in both human and animal research. This would improve overall human health and lower the risk of chronic diseases. Further human studies are required to support the manufacturing of more purple wheat food products and to confirm the physiological advantages of purple wheat products.

## Figures and Tables

**Figure 1 foods-12-01358-f001:**
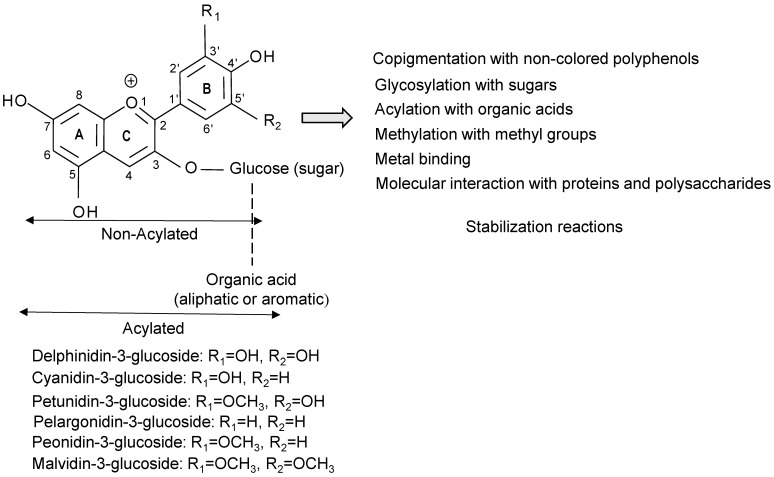
Structures of common anthocyanidins, their anthocyanins-3-glucoside, and possible stabilization reactions.

**Figure 2 foods-12-01358-f002:**
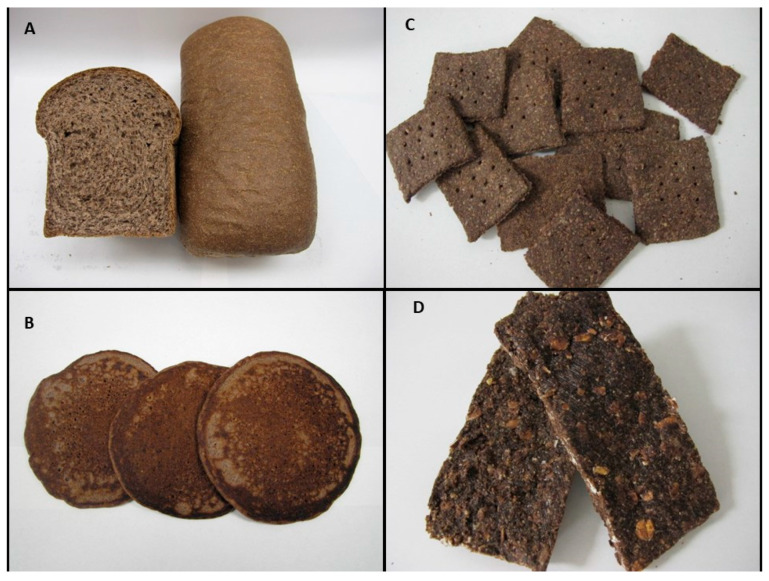
Selected food products made from wholegrain purple wheat: (**A**) Bread, (**B**) Pancakes, (**C**) Bran-enriched crackers, and (**D**) Bran-enriched convenience bars. Crackers and convenience bars were assessed in human studies.

## Data Availability

The data presented in this review are available in the article.

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
