# Peer review of "Purple Wheat: Food Development, Anthocyanin Stability, and Potential Health Benefits"

_foods, 2023, doi:10.3390/foods12071358_

Round 1

Reviewer 1 Report

Gamel et al’s review paper focuses on the health benefits of purple wheat, the effects of processing on anthocyanins, and the processes for stabilizing them. Though the manuscript has good content and structure, there are some issues that should be fully addressed. The following are some of the major issues to which the authors should pay close attention.

It is unclear how the review process was carried out. Major questions remain unanswered, including the type and number of papers targeted, the years of publications covered, and the selection process. These should be briefly highlighted somewhere appropriate by the authors.

As a review paper, the manuscript lacks figures, schemes, tables, and so on. Such illustrations should be used to support some of the authors' descriptions.

The last sentence in section 1 states, "It is the first review that highlights purple wheat food development and its potential in human health," which is incorrect. There are some review papers that highlight the health benefits of purple wheat and its processing (see: Euphytica 56243–258 (1991); Journal of Cereal Science 74, 145-154 (2017), etc).

The structures of the major anthocyanins, as well as the identification process/techniques, should be included where appropriate.

What are the degradation, oxidation, and isomerization products of anthocyanins if wheat processing affects their stability? What effects do the degradation products have on human health? These must be thoroughly explained and supported by the mechanism of degradation/oxidation/ isomerization using representative anthocyanin molecules.

The anthocyanin stabilization processes (described in section 3) such as copigmentation, glycosylation, acylation, encapsulation, and methylation should be illustrated in schemes using representative anthocyanin molecules.

Section 4 should be reorganized. Section 4.2 descriptions should be divided into subsections that describe each benefit (biological activity) one at a time (such as anti-diabetic activity, anti-inflammatory activity, and so on), similar to section 4.1.

There are redundant descriptions all over the place, and the authors should thoroughly revise the manuscript to reduce the number of such descriptions.

Some columns in Table 1 (first page) are cut and should be presented in full.

References should be uniformly listed.

Author Response

Though the manuscript has good content and structure, there are some issues that should be fully addressed. The following are some of the major issues to which the authors should pay close attention.

It is unclear how the review process was carried out. Major questions remain unanswered, including the type and number of papers targeted, the years of publications covered, and the selection process. These should be briefly highlighted somewhere appropriate by the authors.

The review was aimed to cover the purple wheat products development and health impact. It covers all the published studies related to this objective during the last three decades. This information has been included in the introduction.

As a review paper, the manuscript lacks figures, schemes, tables, and so on. Such illustrations should be used to support some of the authors' descriptions.

The manuscript contains an extensive table for the health effect of purple wheat products, which summarized all published in vitro and in vivo animals and human studies on purple wheat products. It also contains the in vitro antioxidants studies. The manuscript also contains a figure for some purple wheat products, which has been extended to include the purple wheat milling fractions.

A figure summarized the stabilization processes for anthocyanins was created and added to the manuscript.

The last sentence in section 1 states, "It is the first review that highlights purple wheat food development and its potential in human health," which is incorrect. There are some review papers that highlight the health benefits of purple wheat and its processing (see: Euphytica 56, 243–258 (1991); Journal of Cereal Science 74, 145-154 (2017), etc).

The sentence is correct. This manuscript focuses on purple wheat products development and health impact. Other reviews focus on different aspects related to purple wheat and/ or anthocyanins, such as genetics and chemistry, for examples:

Euphytica 56, 243–258 (1991): Wheats with purple and blue grains: a review. (This one focus on genetic and breeding). purple and blue grains: a review

Journal of Cereal Science 74, 145-154 (2017): Genetics and chemistry of pigments in wheat grain. (This one focus on genetics and chemistry of anthocyanins).

The structures of the major anthocyanins, as well as the identification process/techniques, should be included where appropriate.

This is not the objective of this review. There are many published reviews described the anthocyanins structure. We have highlighted briefly these reviews, for examples citation [3, 13, 15, 17, 22]

What are the degradation, oxidation, and isomerization products of anthocyanins if wheat processing affects their stability? What effects do the degradation products have on human health? These must be thoroughly explained and supported by the mechanism of degradation/oxidation/ isomerization using representative anthocyanin molecules.

Suggested information was added.

The anthocyanin stabilization processes (described in section 3) such as copigmentation, glycosylation, acylation, encapsulation, and methylation should be illustrated in schemes using representative anthocyanin molecules.

A diagram/ scheme for the anthocyanin stability was added.

Section 4 should be reorganized. Section 4.2 descriptions should be divided into subsections that describe each benefit (biological activity) one at a time (such as anti-diabetic activity, anti-inflammatory activity, and so on), similar to section 4.1.

Due to the limited number of health effect studies on purple wheat, differentiate them based on the biological activities and impact is not applicable. However, section 4 was reorganized into three subsections: Animal studies, Cell culture studies and Human studies.

There are redundant descriptions all over the place, and the authors should thoroughly revise the manuscript to reduce the number of such descriptions.

Irrelevant and repeated information was deleted. The manuscript was significantly revised and edited.  

Some columns in Table 1 (first page) are cut and should be presented in full.

Table format was readjusted. We are sorry for this technical issue due to word software compatibility.

References should be uniformly listed.

References have been checked and style has been adjusted, wherever required. We are sorry for this technical issue due to word software compatibility.

Reviewer 2 Report

I found the article well presented and higly informative. Only very minor typing errors need to be addressed. In detail:

Page 1, Line 2 from bottom: ...content varies widely among...

Page2, Line 4: ... anthocyanins has been ...

Page 2, Line 11 from bottom: due to its contents ...

Page 3, Line 13: ... thus baking affects their ...

Page4, Lines 5-8 and 16-18: They are repetitive, delete one of them

Page 6, Lines 5-6 from bottom: ...filled with amberlite...

Page 7, Line 3 from bottom: write six in letters

Page 8, Line 8 from bottom: ...or non-acid media, are unstable and this influences the color ...

Page 9, Line 4 from bottom: delete "...of anthocianins..., it is already present in the same phrase.

Page 10, Line 6: ...ABTS values have been...

Page 10, Line 12: ..antioxidants than those made from...

Page 11, Line 29: ...consuming purple wheat...

Page 12, Line 15 of Conclusions: ... nutritional quality especially because it is considered...

Bibliography 

Why some citations (e.g. 5, 11, 21, 23, etc.) are in Italics?

Table 1. In the first page of Table 1 the last column is not visible. Please correct.

Author Response

Comments and Suggestions for Authors

I found the article well presented and higly informative. Only very minor typing errors need to be addressed. In detail:

Page 1, Line 2 from bottom: ...content varies widely among...         Changed

Page2, Line 4: ... anthocyanins has been ...    Changed

Page 2, Line 11 from bottom: due to its contents ...    Changed

Page 3, Line 13: ... thus baking affects their ...            Changed

Page4, Lines 5-8 and 16-18: They are repetitive, delete one of them       Repeated information has been deleted

Page 6, Lines 5-6 from bottom: ...filled with amberlite... Changed

Page 7, Line 3 from bottom: write six in letters  Changed

Page 8, Line 8 from bottom: ...or non-acid media, are unstable and this influences the color ... Changed

Page 9, Line 4 from bottom: delete "...of anthocianins..., it is already present in the same phrase.    Changed

Page 10, Line 6: ...ABTS values have been...  Changed

Page 10, Line 12: ..antioxidants than those made from... Changed

Page 11, Line 29: ...consuming purple wheat...   Changed

Page 12, Line 15 of Conclusions: ... nutritional quality especially because it is considered...   Changed

 Bibliography 

Why some citations (e.g. 5, 11, 21, 23, etc.) are in Italics?   References style was adjusted. We are sorry for this technical issue due to the word software.

 Table 1. In the first page of Table 1 the last column is not visible. Please correct.  Table format was adjusted. We are sorry for this technical issue from the word software.

Round 2

Reviewer 1 Report

The authors have addressed some of the concerns raised during the first round of revisions. However, some critical issues raised previously remain unaddressed.

Point 1: The issue of anthocyanin degradation, oxidation, and isomerization PRODUCTS after wheat processing, in particular, is not addressed. This should be highlighted by showing the respective mechanisms, as well as their effects on human health. The authors may refer to Hiemori et al’s article (J. Agric. Food Chem. 2009, 57, 1908-1914) that shows the thermal degradation of rice anthocyanins.

Point 2 The anthocyanin stabilization processes (described in section 3) such as glycosylation, acylation, and methylation should also be addressed. The authors appear to have misunderstood the point. These stabilization processes are chemical reactions and should be depicted in diagrams using representative anthocyanin molecules. What makes a glycosylated or methylated or acetylated anthocyanin more stable than the aglycone? These have to be explained in detail supported by the reaction processes and chemical structures.

Point 3Regarding the incorporation of the structures of the major anthocyanins and their identification techniques, the authors said that it is out of their scope. However, it is still good to provide a highlight to readers in this regards.

Point 4: With only two figures and a table, the review is still deprived of schemes, figures, and tables. Please provide seed structure/picture of purple wheat and other colored varieties where appropriate.  

Point 5: Introduction part: what does three decades mean? Is it 1993-2023? I could not find cited papers before 2000 in the reference list. Please specify!

Author Response

The authors have addressed some of the concerns raised during the first round of revisions. However, some critical issues raised previously remain unaddressed.

Point 1: The issue of anthocyanin degradation, oxidation, and isomerization PRODUCTS after wheat processing, in particular, is not addressed. This should be highlighted by showing the respective mechanisms, as well as their effects on human health. The authors may refer to Hiemori et al’s article (J. Agric. Food Chem. 2009, 57, 1908-1914) that shows the thermal degradation of rice anthocyanins.

I concur with the reviewer regarding the importance of the anthocyanin degradation issue during processing of anthocyanin-pigmented grains, thus the review highlights how to mitigate this issue in Processed Foods in section 3. The review focuses on purple wheat foods and their health effects not on chemistry of anthocyanin degradation in solutions or extracts. The degradation topic would warrant a separate review due to its complexity. The study of Hiemori et al. that investigates the effect of different cookers on anthocyanin loss in black rice is not relevant to the current review. It also investigates degradation of cyanidin 3-glucoside in solution which is beyond the scope and objectives of this publication.

Point 2:  The anthocyanin stabilization processes (described in section 3) such as glycosylation, acylation, and methylation should also be addressed. The authors appear to have misunderstood the point. These stabilization processes are chemical reactions and should be depicted in diagrams using representative anthocyanin molecules. What makes a glycosylated or methylated or acetylated anthocyanin more stable than the aglycone? These have to be explained in detail supported by the reaction processes and chemical structures.

A diagram is added to capture structures of common anthocyanins and possible stabilization reactions (Figure 1), although the focus of this review is on purple wheat foods and their health effects not on chemistry of anthocyanins degradation and stabilization. The objectives of the review are clearly written and highlighted in the title, abstract and throughout the article to talk about purple food product development and potential health benefits. We do understand the review objectives and I believe they are well addressed in the article.

Point 3: Regarding the incorporation of the structures of the major anthocyanins and their identification techniques, the authors said that it is out of their scope. However, it is still good to provide a highlight to readers in this regards.

Structures of the six common anthocyanidins and their anthocyanin derivatives are included in the review (Figure 1) to provide the reader with a sense of anthocyanin structure.

Point 4: With only two figures and a table, the review is still deprived of schemes, figures, and tables. Please provide seed structure/picture of purple wheat and other colored varieties where appropriate. 

Another Figure is added despite the fact that there is no standard number of Tables and Figures required for research or review article. It is all about the content and novel information presented in the article. The review talks about purple wheat foods (Figure 2) and health benefits (Table 1).

Point 5: Introduction part: what does three decades mean? Is it 1993-2023? I could not find cited papers before 2000 in the reference list. Please specify!

The “three decades” is removed and the sentence is rephrased to address the reviewer comment.
